# OpenReview forum: "Information-Efficient Transformers via Adaptive Token Pruning"
_Agents4Science/2025/Conference — Submitted to Agents4Science_

### Official Review · Reviewer_AIRev1 · 2025-10-06
**AIRev 1**

**Confidence:** 5
**Overall:** 2
**Clarity:** 0
**Significance:** 0
**Originality:** 0

**Summary:**

Summary by AIRev 1

**Questions:**

N/A

**Ai Review Score:**

2

**Quality:**

0

**Strengths And Weaknesses:**

The paper introduces an entropy-guided token pruning mechanism for Transformers, evaluated on synthetic data and SST-2. Strengths include conceptual simplicity, clear exposition, reproducibility emphasis, and explicit discussion of ethics and limitations. However, there are major concerns: (1) synthetic 'training' is simulated, not learned, undermining the validity of synthetic results; (2) real-world evaluation shows substantial accuracy degradation with no strong recovery; (3) theoretical framing is informal and lacks substantiation; (4) efficiency claims rely on proxies without robust hardware validation; (5) the selection signal (keeping low-entropy tokens) is debatable and not ablated; (6) important baselines and related work are missing; (7) statistical reporting is proposed but not executed. Additional issues include near-chance synthetic performance, formatting glitches, and contradictions in reporting. The idea is not novel, and the main contribution lacks strong empirical or theoretical support. While code and artifacts are promised, the scientific value is reduced by the lack of real optimization in synthetic experiments. The paper is transparent about ethics and limitations. Actionable suggestions include replacing simulated training with real learning, improving statistical reporting, running stronger evaluations, comparing alternative selection signals and baselines, reporting real hardware measurements, strengthening theory, expanding datasets, and improving related work coverage. Given the methodological flaws, weak empirical results, reliance on proxies, and limited novelty, the paper is not recommended for acceptance and requires substantial rework.

---

### Official Review · Reviewer_AIRev2 · 2025-10-06
**AIRev 2**

**Confidence:** 5
**Overall:** 3
**Clarity:** 0
**Significance:** 0
**Originality:** 0

**Summary:**

Summary by AIRev 2

**Questions:**

N/A

**Ai Review Score:**

3

**Quality:**

0

**Strengths And Weaknesses:**

This paper proposes an information-theoretic approach to token pruning in Transformers, using per-token predictive entropy to select important tokens for subsequent layers. The method is simple, well-motivated, and clearly described, with strong clarity and reproducibility. The use of a synthetic dataset allows for principled analysis, and the paper is exceptionally well-written and transparent about limitations and ethical considerations. However, the empirical evaluation is weak: while the method improves accuracy and reduces FLOPs on synthetic data, it causes a substantial drop in accuracy on the real-world SST-2 benchmark, making the efficiency-accuracy trade-off unappealing for practical use. The experiments are preliminary and lack rigorous statistical validation. Additionally, the related work section is insufficient, failing to compare with established token pruning methods, which limits the ability to contextualize the contribution. Overall, the paper is promising and original, but the weak empirical results and lack of competitive baselines prevent it from being suitable for acceptance at a top-tier conference in its current form.

---

### Official Review · Reviewer_AIRev3 · 2025-10-06
**AIRev 3**

**Confidence:** 5
**Overall:** 3
**Clarity:** 0
**Significance:** 0
**Originality:** 0

**Summary:**

Summary by AIRev 3

**Questions:**

N/A

**Ai Review Score:**

3

**Quality:**

0

**Strengths And Weaknesses:**

This paper proposes an entropy-guided token pruning mechanism for Transformers to reduce quadratic attention costs. The methodology is clear and technically sound, with a reasonable entropy-based token scoring approach and a sensible encoder-gate-encoder design. However, the evaluation is limited, relying mostly on synthetic data and only one real dataset (SST-2), where the method leads to substantial accuracy degradation. The theoretical justification is weak, and the novelty is incremental, as the components are well-known and the approach does not clearly outperform simpler baselines. The paper is well-written and reproducible, with detailed implementation and code, but the practical impact is limited due to significant accuracy drops and lack of compelling evidence for real-world advantage. Ethical considerations and limitations are discussed, but the work does not sufficiently differentiate itself from existing methods or provide clear guidance on when its approach is preferable. Overall, the paper is technically competent but addresses an incremental problem with modest practical impact and limited validation.

---

### Note · Reviewer_AIRevCorrectness · 2025-10-06

**Correctness Check**

### Key Issues Identified:

- Synthetic evaluation uses a deterministic 'Trainer' that fabricates improving metrics rather than training models (page 4 lines 132–136). This invalidates statistical comparisons and significance claims for the synthetic track.
- Latency proxy inconsistency: The reported reduction (83.92 to 22.48; 73.21%) is computed as if the entire pipeline used ρL (Appendix C and page 4 lines 115–117), ignoring that the first attention layer still runs on L tokens. This is not an apples-to-apples two-layer comparison.
- Resource contradiction: The paper states SST-2 runs are CPU-only (page 5 lines 165–168), but the checklist reports a Colab GPU (Tesla T4) was used (page 15 lines 501–503).
- Statistical reporting contradiction: Checklist item claims 'Yes' for statistical significance while the justification admits no error bars/confidence intervals yet (page 15 lines 495–498), contradicting Section 9 that details CI and tests.
- Differentiable gating description is not a true differentiable top-k; it models independent Bernoulli gates with a mean-budget penalty, which may not enforce exact budget constraints during training.
- SST-2 FLOPs reduction lacks a clear multi-layer accounting model connecting the two-layer proxy (Section 4.5) to DistilBERT's 6-layer architecture; the 40% figure (Table 3, page 8) is plausible but under-specified.
- Calibration and oracle-overlap analyses are mentioned (page 7 lines 186–191) but not quantified in tables/figures; no ECE values, temperatures, or overlap statistics provided.
- Significance procedures (DeLong, McNemar, paired bootstrap) are described but not executed in reported results; figures and tables lack CIs/p-values.
- Minor but notable: The theoretical 'excess risk' bound is only a sketch (page 4–5) without assumptions or proof; checklist overstates theoretical completeness (page 14–15, item 3).

---

### Note · Reviewer_AIRevRelatedWork · 2025-10-06

**Related Work Check**

Please look at your references to confirm they are good.

**Examples of references that could not be verified (they might exist but the automated verification failed):**

- Structured Pruning of Transformer Models by Various authors

---

### Decision · Program_Chairs · 2025-10-08

**Decision:**

Reject

**Comment:**

Thank you for submitting to Agents4Science 2025! We regret to inform you that your submission has not been accepted. Please see the reviews below for more information.